# Exploring the Support Needs of Chinese Family Carers of People Living with Dementia in New Zealand during the COVID-19 Pandemic: A Resilience Resources Framework Perspective

**DOI:** 10.3390/ijerph21070946

**Published:** 2024-07-19

**Authors:** Fei Li, John Parsons, Gary Cheung

**Affiliations:** 1School of Nursing, Faculty of Medical and Health Sciences, University of Auckland, Auckland 1142, New Zealand; j.parsons@auckland.ac.nz; 2School of Nursing, Guangxi Medical University, Nanning 530021, China; 3Department of Psychological Medicine, School of Medicine, Faculty of Medical and Health Sciences, University of Auckland, Auckland 1142, New Zealand; g.cheung@auckland.ac.nz

**Keywords:** support needs, family carer, COVID-19, Chinese, dementia, resilience resources

## Abstract

Caring for people living with dementia during the novel coronavirus disease 2019 (COVID-19) pandemic significantly impacted the emotional, physical, and social well-being of carers. However, no study has focused on the well-being of Chinese carers of people living with dementia in New Zealand during the pandemic. This study aimed to explore the support needs of Chinese carers of people living with dementia in New Zealand during the COVID-19 pandemic. Semi-structured interviews were conducted by two bilingual and bicultural researchers. Thematic analysis was used to explore the resilience resources for Chinese carers. Twelve Chinese carers were recruited from four community organizations in New Zealand. Four themes were identified: (1) social isolation, (2) emotional loneliness, (3) ambivalent feelings of being a carer, and (4) a variety of unmet needs. The findings of our study provide new insights into the multiple support needs of Chinese carers of people living with dementia during the COVID-19 pandemic. Implications for practice include the establishment of culturally appropriate care support services and the development of tailored resilience-building interventions to address the unmet needs of Chinese carers of people living with dementia.

## 1. Introduction

Being a carer for people living with dementia is a stressful experience. However, the COVID-19 pandemic exacerbated these negative effects on family carers’ mental well-being [1]. A cross-sectional study of 320 people living with dementia and their family carers in Argentina, Brazil, and Chile explored the impacts of social isolation. The results showed that family carers experienced increased feelings of fatigue and being overwhelmed during the COVID-19 pandemic [2]. In Portugal, family carers of people with a neurocognitive disorder experienced an increase in caring burden and a decline in their well-being after COVID-19 home confinement [3]. According to these collective studies, family carers were particularly vulnerable and isolated during the pandemic.

The number of people living with dementia is increasing in New Zealand, and its Asian population is growing faster than those of the other main ethnic groups. As the Chinese population is the largest Asian ethnic group in New Zealand (4.9% of the total population) [4], the number of Chinese people living with dementia is expected to increase. Several studies have explored the lived experiences of people living with dementia and their families among New Zealand’s ethnic groups, including Chinese [5], Indian [6], and Māori [7]. These qualitative studies consistently highlighted the importance of addressing the culturally sensitive needs of these ethnic groups. Key dementia services in New Zealand are provided by member organizations of Alzheimer’s New Zealand and Dementia New Zealand [8]. These services are primarily designed for the mainstream European population and typically include dementia advice, carer support, carer education and living well with dementia services. However, there is a lack of specific support systems for people from diverse cultural backgrounds, which could be a barrier for carers who need cultural or language-specific resources and support. However, little research has been conducted to explore the support needs of dementia family carers during the COVID-19 pandemic in New Zealand.

There is evidence demonstrating that family carers of people living with dementia retain resilience before and during the COVID-19 pandemic [9,10]. However, resilience is a multidimensional response to the caring role and is influenced by a wide range of interrelated factors [11]. A UK study applied an ecological model of resilience to explore the experiences of eight people living with dementia and 42 unpaid carers during the COVID-19 pandemic [9]. This study identified organizations and social support services that could be considered resilience factors for people living with dementia and their carers. Although this framework has been used to identify factors that support informal carers of people living with dementia [9,12,13] and Chinese families during the COVID-19 pandemic [14], it has not been utilized to explore existing resilience resources to support Chinese carers of people living with dementia. Therefore, it remains unclear whether the resilience resources identified in previous studies adequately capture the support needs required for Chinese carers of people living with dementia in New Zealand. 

To capture the resilience resources of Chinese carers of people living with dementia, this study was guided by the resilience resources framework [13]. This framework was adopted from the ecological resilience framework developed by Windle and Bennett (2011) [15], which has been widely used in various studies. However, this framework is not specific to the experience of dementia [16]. Therefore, Han and colleagues modified this framework for carers of people living with dementia, providing valuable insights into the resilience resources that support people with dementia and their carers, particularly in the context of hospice care [13]. Although there is no standard definition of resilience resources in dementia carers research, a common understanding is that resilience is shaped by a variety of resources, including individual resources, community resources, and societal resources. Therefore, the operational definition of resilience resources in this study can be defined as carers holding these resources to facilitate or hinder resilience. The resilience resources that family carers possess or lack that either promote or hinder their ability to cope effectively with the challenges of caring for someone with dementia. The application of the resilience resources framework can advance knowledge of the resilience of Chinese family carers of people living with dementia.

The resilience resources framework [13] was used to guide the data analysis by exploring the present and absent resilience resources among Chinese carers of people living with dementia. By exploring resilience resources, this study aimed to understand the support needs of Chinese carers of people with dementia during the COVID-19 pandemic in the New Zealand context. If carers are deficient in these resilience resources, this may lead to compromised well-being and increased difficulties in their caring responsibilities. To address these challenges, it is important to understand and link their support needs to the availability of resilience resources. The study findings may facilitate the development of resilience-building interventions or service improvements to strengthen their caring capacity and enable them to cope effectively with future challenges in a similar pandemic in the future.

## 2. Materials and Methods

### 2.1. Design

A descriptive qualitative method was used to understand the support needs of Chinese family carers of people living with dementia during the COVID-19 pandemic. The rationale for using this method was to understand the individual human experience and perception in its unique context [17]. From January 2022 to June 2022, we conducted a qualitative study using semi-structured interviews with Chinese carers of people living with dementia. Impacted by the COVID-19 physical restrictions, all interviews were conducted by telephone. 

### 2.2. Setting and Participants

Purposive sampling was used in this study to identify those participants who were most likely to provide abundant information about the phenomena under study [18]. The inclusion criteria were age 18 years and above, provision of care or support to a Chinese person living with dementia, ability to communicate in Mandarin or Cantonese, and ability to provide informed consent. The exclusion criteria were verbal communication disorders or hearing disorders. 

Potential participants were recruited from four non-profit communities (Age Concern Auckland, Dementia Auckland, Dementia Wellington, and Dementia Canterbury). These organizations were selected because they are the primary providers of dementia services in the three largest cities of New Zealand, where most of Chinese New Zealanders live. The first author contacted those who had been referred to determine their eligibility, provide information, answer questions, set up interviews, and obtain written informed consent.

### 2.3. Data Collection

Two female and bilingual (Cantonese/Mandarin) researchers with a nursing background conducted the telephone interviews. They had no existing relationship with the participants or the four non-profit organizations. The interviews were conducted following a semi-structured guide developed by the research team (Appendix A). All interviews were audio-recorded and transcribed verbatim by the two bilingual and bicultural researchers.

### 2.4. Data Analysis

Data analysis was conducted as soon as data collection was completed. Thematic analysis is an appropriate qualitative method that is used for data analysis in small-scale studies that collect data from semi-structured interviews [19]. In this study, both deductive and inductive approaches were used in data analysis to explore the themes. A deductive approach was based on a pre-existing framework and guided by the research questions [20]. In contrast, an inductive approach was based on a systematic procedure for analyzing qualitative data and guided by specific objectives [21]. A ‘bottom-up’ inductive approach was adopted to explore the emergent themes. The themes were identified at a ‘semantic’ level from the transcripts, based on the surface-level meaning of the data and without exploring the underlying meaning of participants’ words. The deductive approach was used to map the identified themes onto three levels within the framework of resilience resources. The entire research team participated in the data analysis procedure.

The data analysis process followed a 6-phase guide proposed by Braun and Clarke (2006) [20]. Phase 1 was to become familiar with the data by reading the transcripts. Phase 2 was to generate initial codes by coding the data line by line using Microsoft Excel and Word. Phase 3 was to generate themes based on the codes. Phase 4 was to review, modify, and develop the preliminary themes. This preliminary analysis was discussed with the second author before being refined and organized into broad themes. Phase 5 was to name and define the themes. The research team reviewed and iteratively agreed on the resultant themes over five meetings. Phase 6 was to write the report [22]. 

### 2.5. Rigor

The rigor of this study was based on the principle of trustworthiness developed by Lincoln and Guba (1985), which includes credibility, confirmability, dependability and transferability [23]. To enhance the credibility and confirmability, the first author met frequently with other authors experienced in qualitative research and carer research to discuss the study design, formulate the analysis strategy, and receive ongoing feedback on preliminary findings. To ensure dependability of our findings, two interviewers used the same interview guide. After interviewing each participant, the verbatim manuscripts were analyzed and discussed with the other authors to minimize errors in the analysis. Finally, participants were recruited by a single researcher to increase transferability. In addition, the entire interview was tape-recorded in order to reduce the mission of the data.

### 2.6. Ethical Considerations

Ethical approval was obtained (Reference Number AH23674). Written informed consent was obtained before the interview.

## 3. Results

Thirteen family carers were referred by the community organizations, and twelve family carers of people living with dementia provided informed consent and completed the telephone interviews. All family carers were female. Most family carers were older than 65 years (66.7%) and were wives (60%). Ten carers reported having at least one health condition, including knee problems (*n* = 3), depression (*n* = 3), lumbar vertebral disease (*n* = 2), hypertension (*n* = 2), heart disease (*n* = 1), cancer (*n* = 1), psoriasis (*n* = 1), hypercholesterolemia (*n* = 1), and long COVID-19 fatigue (*n* = 1). The characteristics of the family carers are summarized in Table 1. The interviews lasted between 38 and 87 minutes.

Four themes were generated from the data. The themes and sub-themes identified were mapped onto the resilience resources framework. Table 2 provides an overview of the presence and absence of resilience resources for Chinese carers of people living with dementia during the pandemic at different ecological levels. The presence of resilience resources focused on the themes or sub-themes that show how existing resources contribute to the resilience of family carers. Coping strategies and family responsibilities were the only existing individual resilience resources among Chinese family carers of people living with dementia. There were no existing community and societal resilience resources. The absence of resilience resources focused on themes or sub-themes that illustrate the negative impact of the absence of resilience resources on family carers. Most of the themes and sub-themes presented in the absence of resilience resources include those at the individual, community, and societal levels.

In addition, certain themes and sub-themes appear to be relevant across multiple ecological levels. For example, emotional loneliness was found to be associated with both individual and community levels. Similarly, a lack of awareness of local dementia services was identified as a challenge at both individual and societal levels. In addition, the lack of culturally appropriate services and the suspension of social services during the COVID-19 pandemic were identified as overlapping issues at the community and societal levels. 

### 3.1. Theme: Social Isolation

All carers expressed feelings of social isolation when their relatives or friends were unable to visit them during the COVID-19 pandemic. However, this feeling was more pronounced among carers over the age of 65. Many carers expressed fear of contracting the virus, which prevented them from participating in social or support activities. Some expressed reluctance to go to public places other than supermarkets, and some continued not to go out despite the lifting of physical distance restrictions. The COVID-19 pandemic had a significant impact on their lifestyle.


*Before COVID-19, we had many friends who came to visit us, but now no one comes to visit us. We had no social activities after COVID-19. My husband and I have not left the house for almost a year. Being aware of our vulnerability as elderly people, we have become accustomed to staying at home. Despite the easing of social restrictions, we have developed the habit of staying indoors.*
Carer (wife), Interview 8

### 3.2. Theme: Emotional Loneliness 

Ten carers expressed a feeling of emotional loneliness. Some of them mentioned the difficulty of sharing their feelings with friends or children who could not empathize with their situation. Furthermore, the social restrictions and resulting isolation during the lockdown period may have exacerbated their feelings, as they had limited emotional support available to them.


*Instead of seeking comfort from my friends, I preferred to handle these issues on my own. Being an immigrant, I don’t have many friends in New Zealand… There is no use in talking to my friend when she cannot provide any useful suggestions.*
Carer (daughter), Interview 11

### 3.3. Theme: A Variety of Unmet Needs 

A variety of unmet needs were identified in: (1) suspension of social services during the COVID-19 pandemic, (2) lack of dementia knowledge and skills, (3) lack of awareness about local dementia services, and (4) lack of culturally appropriate services. 

#### 3.3.1. Sub-Theme: Suspension of Social Services during the COVID-19 Pandemic

The pandemic resulted in the suspension of numerous social support services. Carers expressed a variety of needs for assistance, including house cleaning (five carers), respite services, transport services (three carers), and volunteer visiting services (two carers). Furthermore, a few spouse carers reported instances of physical aggression from their husbands, adding another layer of complexity to their caring role.


*Before the lockdown, we had a volunteer who visited and talked to my husband once a week. Sometimes we even went out together for "Yum Cha"... Sometimes we went for a walk… Since the lockdown began, the volunteer has not been able to come, and we haven’t seen or heard from them since.*
Carer (wife), Interview 6

#### 3.3.2. Sub-Theme: Lack of Dementia Knowledge and Skills

Four carers expressed a strong desire to gain further knowledge about dementia care to adequately prepare for their loved one’s cognitive decline. 


*To maintain my friend’s memory, her daughter always said ‘I will do anything to make my mother happy’. While I considered this approach to be overly simplistic, I was uncertain if there might be a better option available.*
Carer (friend), Interview 12

#### 3.3.3. Sub-Theme: Lack of Awareness about Local Dementia Services 

Chinese carers faced challenges due to language barriers, resulting in a lack of resources and community connections. They were often unaware of the dementia services available for people living with dementia in New Zealand, even if they were English-speaking. Despite this, they still struggled to find community services. In addition, when they reached a point where they could no longer care for themselves at home, they sought information about residential care. They also struggled to find respite care services and did not know where to start.


*As immigrants, this is our first experience caring for a person with dementia. We are unsure about where to seek information or what services are available to us. If my mother-in-law’s condition deteriorates and we are unable to provide the necessary care at home, will the government provide any assistance? I have no idea where I can obtain additional information on this matter.*
Carer (daughter-in-law), Interview 5

#### 3.3.4. Sub-Theme: Lack of Culturally Appropriate Services 

Existing dementia services were not culturally appropriate for Chinese people living with dementia and their carers. Some carers had access to dementia support services, but they complained that the services were unsuitable for Chinese people living with dementia. 


*In 2019, he participated in a community centre programme that was specifically designed for older people. The programme offered numerous activities, including singing and playing games, and free morning tea and lunch. However, he later refused to go because he felt uncomfortable with the European food at the centre…Several facilities have been designed for Europeans, but there is no Chinese community centre.*
Carer (wife), Interview 2

### 3.4. Theme: Ambivalence of Being a Carer

Three subthemes were developed in this theme: (1) negative feelings, (2) family obligations—marital ties and filial piety, and (3) coping strategies.

#### 3.4.1. Sub-Theme: Negative Feelings

Nine carers experienced a range of negative feelings during the COVID-19 pandemic, including stress, sadness, burden, fatigue, upset, desperation, exhaustion, intolerance, distress, annoyance, and anger. The majority of them reported feeling stressed due to the increased challenges of caring for someone with dementia-related behavioral and memory impairments without additional support during the lockdown. 


*I feel unhappy because I have to stay at home all the time… Sometimes I’m depressed because of this ongoing pandemic… Two days ago, I heard about the level-three and red traffic lights from the news. I had hoped that the situation would improve and that we would soon be able to resume normal activities, but the situation has become urgent again…so my unhappiness has grown.*
Carer (wife), Interview 6

#### 3.4.2. Sub-Theme: Family Obligations—Marital Ties and Filial Piety 

Chinese carers bring traditional Chinese values and beliefs from their home country to New Zealand. When it comes to traditional gender roles and family values, some spouse carers have expressed the irreplaceable role of caring within their families. The strong marital ties further motivate them to continue in their caring roles.


*My friends and family all noted how difficult it was for me to care for my husband. They often suggest, ‘Let us take him to the nursing home’. However, I simply cannot bear to let him go. Not right now, until he couldn’t remember anything about me…We’ve been married for 54 years, and my husband has expressed his belief that we still have another 20 years together.*
Carer (wife), Interview 4

Influenced by traditional Chinese filial piety, children take responsibility for caring for their parents.


*Partially obligation you know, because I’m the oldest. I feel it is my role…My younger sister used to look after my parents. You know, my father passed away three years ago, but she’s done here with our father. And now, it’s my turn to look after my mother.*
Carer (daughter), Interview 3

*I’m the only child in my family… I will be the only one my parents will depend on in the future*.Carer (daughter), Interview 11

#### 3.4.3. Sub-Theme: Coping Strategies

Ten carers spoke about their ability to manage their negative feelings. They employed various strategies to distract themselves from the burden of caring, such as enjoying watching movies and television series or developing new hobbies like cooking. 


*I’ve realised in recent years that you have to do that you have to do things both when you’re happy and when you’re unhappy, right? You could only have an easy day if you maintained a better mood. I always found ways to keep myself busy so that I could get through the day.*
Carer (wife), Interview 1

## 4. Discussion

To the best of our knowledge, this is the first study to explore the support needs of Chinese family carers of people living with dementia in New Zealand during the COVID-19 pandemic. The participants described four themes: social isolation, emotional loneliness, a variety of unmet needs, and ambivalent feelings about being a carer. These findings highlight the impact of culture and the COVID-19 pandemic on the challenging role of a carer, as well as the ongoing lack of support and knowledge in dementia. In addition, the lack of resilience resources at both community and societal levels poses a significant challenge for carers of people with dementia. In such situations, carers may struggle to find adequate support and resources to cope effectively with the demands of their role. This can lead to increased stress, burnout, and reduced quality of life for both carers and people with dementia.

The available evidence indicates that carers of people living with dementia experienced a lack of adequate organizational and community resilience resources during the COVID-19 pandemic, which is consistent with previous findings that the pandemic had a significant impact on the utilization of social support among carers of people living with dementia [24]. Some service providers have offered remote support services (e.g., videoconferences, telephone helplines) as a response to the pandemic, but these services have not been accessible to everyone, particularly older individuals who may not be familiar with technology or lack internet access. In addition, language has consistently been identified as a major barrier to seeking and accessing dementia-related services in other studies involving Chinese populations [25,26]. These findings emphasize the importance of collaborative efforts among various community stakeholders to provide more language- and technology-friendly dementia services.

Emotional loneliness was the theme most frequently noted by carers. A pre-pandemic qualitative study in New Zealand reported that Chinese late-life immigrants expressed a sense of loneliness [27]. Chinese late-life immigrants were unable to develop their connections and social networks within the local community because they adhered to traditional Chinese beliefs and values. As in-person support services closed during the lockdown, online culturally supportive activities and services were not available for Chinese carers of people living with dementia to interact socially with other carers. As a result, their feelings of emotional isolation were sustained or exacerbated during the pandemic. These findings also highlight the potential for developing peer support activities to improve the mental well-being of carers of people living with dementia by enabling social cohesion. 

Carers reported a wide range of unmet needs during the pandemic. This is likely because the experiences of living with dementia differ greatly among individuals [28]. Thus, the experiences of those caring for people living with dementia vary widely. This finding was consistent with the results of a recent systematic review that identified and synthesized the existing literature on the needs of family carers of people living with dementia at home and found that unmet needs varied [29]. Although some of these unmet needs may not be lockdown-dependent, they are the common needs reported by previous studies. For example, the lack of dementia knowledge is often described in the literature [30,31]. Given that the pandemic may not be an ongoing issue, these non-COVID-dependent unmet needs could be addressed by organizations working closely with carers to understand their support needs.

It is noteworthy that our data collection was conducted from January to June 2022, as New Zealand transitioned from Red to Orange under the COVID-19 Protection Framework (traffic light system). The COVID-19 Protection Framework was announced with the country placed at Red on 23 January 2022, shifting to Orange on 13 April 2022. Under the Orange traffic light setting, people are instructed to wear a mask in most indoor settings but are not limited to indoor and outdoor gatherings. There was a greater easing of restrictions when the traffic light was set to Orange than when the traffic light was set to Red. With the removal of some COVID-related restrictions, social isolation and unmet needs are likely to change. However, feelings of emotional loneliness may not be lockdown-specific and may persist after the restrictions are lifted. This emphasizes the value of ongoing research to explore the unique needs of Chinese family carers after COVID-19.

All carers mentioned that they were able to cope with the challenges of providing care during the COVID-19 pandemic. This finding aligned with a prior qualitative study on the experiences of living with dementia in Chinese New Zealanders, which suggested that Chinese carers of people living with dementia employed a range of strategies to maintain their well-being [5]. As reported by many carers, digital products (telephone, computer, or iPad) were the most important tools for them during the lockdown. It is worth mentioning that digital products are valued for social connections with family or friends [32]. Some carers reported no physical visits with friends and family, which made them feel bored and lonely. Using digital products to stay connected with family or friends was a source of comfort. Overall, the ability to use digital products can be regarded as a reliable source of resilience for carers [33].

Our findings demonstrated the essential role of resilience resources in supporting the well-being of family carers during the COVID-19 pandemic. We acknowledged that the COVID-19 pandemic and caring for people living with dementia are the main challenges for family carers. These challenges can be mitigated by enhancing the carer’s resilience resources. Notably, one-third of absence resilience resources can be interpreted at both individual and community levels, community and societal levels, or individual and societal levels. For example, emotional loneliness was found to be related to both the individual and community levels. Emotional loneliness was mapped to the individual level because carers felt that caring for people living with dementia was a lonely journey. Furthermore, emotional loneliness was mapped to the community level because these carers received limited emotional support from their family and friends during the COVID-19 pandemic. This is in line with the findings from a previous study [13], which identified a few overlapping resources at both the community and societal levels. This is because the purpose of the framework of resilience resources was to categorize these support needs across the ecological levels rather than differentiate them at each level [13].

### 4.1. Implications of the Study

The primary implication of this study is to inform the development of resilience-enhancing interventions that can strengthen existing resilience resources and tailor the intervention to the preferences and needs of family carers. The recommendations for utilizing these resilience resources are as follows. First, a need-driven and culturally appropriate psychoeducation intervention at the individual level should (1) provide culturally appropriate information or guidance in the context of the dementia care journey and community dementia services; (2) train family carers to manage dementia symptoms and develop communication skills and coping strategies to maintain positive emotions; and (3) be tailored to the needs of Chinese carers. Second, interventions at the community level should provide specific strategies to support family carers where their voices can be heard and to provide emotional support through the caring journey, for example, an online support group for carers with no access to a vehicle. Third, interventions at the societal level should involve collaboration with the community and societal stakeholders to raise awareness of dementia in society. In addition, existing societal services should account for the needs of users from a variety of cultural backgrounds to ensure that carers have access to the appropriate support while also addressing inequalities for New Zealand’s minority population. Finally, our findings emphasize the importance of integrating resources from the individual, community, and societal levels to support carers of people living with dementia in developing their resilience system. 

### 4.2. Limitations of the Study

This study has several limitations. First, this study had a small sample size. However, given the relatively specific focus of the study, the criteria for purposive sampling, and the richness of the data generated, the sample size was considered sufficient to achieve our objective. Some of our findings were consistent with those previous studies, such as family obligations for being a carer [5,34]. Second, the generalizability of the findings could be limited by the fact that the participants were recruited from two cities in New Zealand. Third, there is a lack of diversity in the sample of carers because most carers were wives and older people. Future research is needed to include a larger sample with diverse characteristics, such as male carers, different speakers of Chinese dialects, and people living with dementia transitioning to living in long-term care settings. 

## 5. Conclusions

Our findings indicate that various support needs mentioned by Chinese carers of people living with dementia existed prior to COVID-19 and were exacerbated during the COVID-19 pandemic. In addition, our study identified both resource gaps and resilience resources among Chinese carers during the COVID-19 pandemic in New Zealand. These findings have important implications for the development of targeted interventions aimed at enhancing resilience and the provision of culturally sensitive services for carers of people living with dementia, particularly for those with limited social networks, technological skills, or English proficiency.

## Figures and Tables

**Table 1 ijerph-21-00946-t001:** Characteristics of family carers (*n* = 12).

Characteristics		Number of Family Carers	Percentage (%)
Age			
	<65 years	4	33.3
	≥65 years	8	66.7
Gender			
	Female	12	100.0
Birthplace			
	Hong Kong	6	50.0
	Mainland China	3	25.0
	Singapore	1	8.3
	Taiwan	1	8.3
	New Zealand	1	8.3
Relationship with care recipients			
	Wife	8	66.7
	Daughter	2	16.7
	Daughter-in-law	1	8.3
	Friend	1	8.3
Years of caring			
	<1 year	5	41.7
	1–3 years	2	16.6
	4 years and over	5	41.7
Living location			
	Auckland	11	91.7
	Christchurch	1	8.3
Health conditions			
	Chronic diseases	8	66.7
	Depression	3	25.0

**Table 2 ijerph-21-00946-t002:** Summary of the presence and absence of resilience resources among Chinese carers of people living with dementia during the COVID-19 pandemic.

Ecological Levels of Resilience Resources	Presence of Resilience Resources	Absence of Resilience Resources
Individual level	Coping strategies; family obligations.	Negative feelings;
		Emotional loneliness;
		Lack of dementia knowledge and skills;
		Lack of awareness about local dementia services.
Community level	N/A	Emotional loneliness;
		Lack of culturally appropriate services;
		Suspension of social services during the COVID-19 pandemic.
Societal level	N/A	Social isolation;
		Lack of awareness about local dementia services;
		Lack of culturally appropriate services;
		Suspension of social services during the COVID-19 pandemic.

Note: N/A-Not Applicable.

## Data Availability

In order to protect the privacy of individual participants, data sharing is not applicable to this article.

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
