# Peer review of "Exploring the Support Needs of Chinese Family Carers of People Living with Dementia in New Zealand during the COVID-19 Pandemic: A Resilience Resources Framework Perspective"

_ijerph, 2024, doi:10.3390/ijerph21070946_

Round 1
Reviewer 1 Report
Comments and Suggestions for Authors
See attached file.

Author Response
|
Comments 1: Line 14:in the section on “This study aimed to under the support” the term under should be understand |
|
Response 1: Thank you for pointing this out. We agree with this comment. Therefore, we have revised the manuscript and this change can be found on - page 2, paragraph 4, line 78. |
|
Comments 2: Line 49: dementia family carers – edit as family carers of people with dementia |
|
This change can be found on – page 2, paragraph 2, line 52. |
|
Comments 3: Line 172: Table 2. Summary of resilience resources among Chinese carers of people – include presence and absence of resilience resources into the title |
|
Response 3: We agree with this comment. Therefore, we have changed the title of Table 2 into “Summary of the presence and absence of resilience resources among Chinese carers of people – include presence and absence of resilience resources”. |
|
Comments 4: Line 180: presence or absence - should be presence and absence |
|
Response 4: We have revised it. This change can be found on – page 5, paragraph 1, line 190. |
|
Comments 5: Line 312 –first paragraph of discussion. The themes are not inclusive of the presence of resilience resources. Need to address in this discussion introduction (need to comments on presence of resources among two levels of community and societal level). |
|
Response 2: We have revised this paragraph. Comments on current resources between two levels of community and society have been added to the manuscript.
This information is provided in the manuscript– page 8, lines 358-362. “In addition, the lack of resilience resources at both community and societal levels poses a significant challenge for carers of people with dementia. In such situations, carers may struggle to find adequate support and resources to cope effectively with the demands of their role. This can lead to increased stress, burnout and reduced quality of life for both carers and people with dementia.” |
|
Comments 6: Review the manuscript to make sure reference numbers in the text are superscript (noted that some are not). |
|
Response 6: The reference has been thoroughly checked. |
Reviewer 2 Report
Comments and Suggestions for Authors
Change of description format
P2 L48  delete
P3 There are unnecessary characters on the left side of the table
P3 L171 If row 171 is deleted, row 174 on page 5 can be moved and the table2 can be written on page 1.
P7    Table delete
P8 L321 dementia carers 20 → 20 ?
P8 L342  among individuals 24 → 24?
P10 L452   Unable to check appendix
Comment
1.What support systems does New Zealand have for people from other cultures? I think it is necessary to provide an overview.
2.How were the subjects selected from the Chinese population, who make up 4.9% of New Zealand's population?
It is necessary to describe include the method for selecting the four facilities.
3.I would like to know a description of the especially support that was necessary during the coronavirus pandemic period, including a comparison with normal support.
Author Response
|
3.1 General suggestions |
|
P2 L48  delete |
|
Response : We acknowledge these errors and have addressed them accordingly. |
|
P3 There are unnecessary characters on the left side of the table |
|
Response : We acknowledge these errors and have addressed them accordingly. |
|
P3 L171 If row 171 is deleted, row 174 on page 5 can be moved and the table2 can be written on page 1. |
|
Response : We acknowledge these errors and have addressed them accordingly. |
|
P7 Table delete |
|
Response : We acknowledge these errors and have addressed them accordingly. |
|
P8 L321 dementia carers 20 → 20 ? |
|
Response : I acknowledge this formatting error, which has now been amended. |
|
P8 L342  among individuals 24 → 24? |
|
Response : I acknowledge this formatting error, which has now been amended. |
|
P10 L452   Unable to check appendix |
|
Response : The appendix file has been reuploaded. |
|
|
|
3.2 Specific comments |
|
Comments 1: What support systems does New Zealand have for people from other cultures? I think it is necessary to provide an overview. |
|
Response 1: Thanks for raising this issue. We agree with this comment. We have accordingly added an overview to emphasize this point.
This overview information is provided in the background (Page number 2, Paragraph 1, Lines 45-47). “Key dementia services in New Zealand are provided by member organisations of Alzheimers New Zealand and Dementia New Zealand (Weir, 2019). These services are primarily designed for mainstream European population and typically include dementia advice, carer support, carer education and living well with dementia services. However, there is a lack of specific support systems for people from diverse cultural backgrounds, which could be a barrier for carers who need cultural or language specific resources and support”. Weir, A. (2019). Effective Restorative Home Support for Older People Living with Dementia and Their Caregivers: A New Zealand Case Study. IntechOpen. doi: 10.5772/intechopen.73165 |
|
Comments 2: How were the subjects selected from the Chinese population, who make up 4.9% of New Zealand's population? |
|
Response 2: We worked with four dementia and community organisations to identify and recruit Chinese participants. This purposive sampling was used because we don’t have access to the contact details of the general Chinese population that make up 4.9% of the New Zealand’s population. The limitations of purposive sampling have implications for generalising our study findings and we have already included this as a limitation in the initial submission. |
|
Comments 3: It is necessary to describe include the method for selecting the four facilities. |
|
Response 3: Thank you for your suggestions. Age Concern Auckland, Dementia Auckland, Dementia Wellington, and Dementia Canterbury were selected because they are the primary providers of dementia services in the three largest cities of New Zealand where most of Chinese New Zealanders live. This overview information is added to Section 2.2. Setting and participants--page number 2 and line 111-113. “These organisations were selected because they are the primary providers of dementia services in the three largest cities of New Zealand where most of Chinese New Zealanders live.” |
|
Comments 4: I would like to know a description of the especially support that was necessary during the coronavirus pandemic period, including a comparison with normal support. |
|
Response 4: We are unclear about this comment. Do you mean support that was necessary for Chinese people with dementia during the COVID-19 pandemic when compared with support for the general population? If we have interpreted your comment correctly, there is no data for such comparison. However, we have provided some information of increased support for older people during the pandemic.
During the COVID-19 pandemic, there were examples of increased support for people living with dementia and their carers. For example, the interRAI-Contact Assessment (interRAI-CA) was used as a brief assessment tool to triage the support needs of adults living at home, including people with dementia. The interRAI-CA determines whether the individual requires long-term support services while living at home (i.e., household management and/or personal care), or whether their needs are too high to be met in the community and residential care is required. New Zealand Government agencies, not-for-profit community organisations and academics also joined forces to develop strengths-based messages and interventions to address the psychological and emotional needs, social connection and social recognition of older adults.
Healthcare providers also used innovative ways to provide services to older adults during the lockdown period. For example, dementia community support services used videoconferencing to deliver evidence-based group treatment to people with dementia, providing social contact for them and their carers.
Compared to normal support, healthcare providers also used innovative ways to provide services to older adults during the lockdown period. For example, dementia community support services used videoconferencing to deliver evidence-based group treatment to people with dementia, providing social contact for them and their carers. |

Reviewer 3 Report
Comments and Suggestions for Authors
1. Organizational and social support services are kind of resilience resources and generalized to be the support needs for people living with dementia and their carers. The incentives to conduct this research described in lines 50-61 seemed too weak.
2. Reference 12 is identified factors that support people living with dementia and their carers. Why used this study to be the resilience resources framework?
3. Design: A descriptive qualitative method was used to explore the support needs of Chinese family carers of people living with dementia during the COVID-19 pandemic. The rationale for using this method was to understand the individual human experience in its unique context. There are conflicts between lines 65-70 and lines 75-79.
4. The process of thematic analysis described in lines 117-118 is an inductive approach. Where is the deductive approach in the data analysis?
5. The rigor of this research must be described.
6. The health conditions of the carers might be the confounding factors of the results.
7. The definition of resilience resources has to be stated clearly.
8. What are the main findings of this research? The contents of Table 2 are not consistent with the text.
9. The authors need to reorganize the research’s findings into one Table.
10. What are the meanings of Title1-Title4?
11. What are the differences between support needs and resilience resources?
12. Discussion: “The available evidence indicates that dementia carers experienced a lack of adequate organizational and community support services during the COVID-19 pandemic, which is consistent with previous findings that the pandemic had a significant impact on the…. “These results are not seen in the text.
13. lines 378-380: Notably, one-third of absence resilience resources can be interpreted at both individual and community levels, community and societal levels, or individual and societal levels.
lines 386-388: This is because the purpose of the framework of resilience resources was to categorize these support needs across the ecological levels rather than differentiate them at each level 12.
These results are not seen in the text.
Author Response
|
3.1 General suggestions |
|
Are all the cited references relevant to the research? |
|
Response: We have acknowledged that there have been some formatting errors. All references have been carefully checked. |
|
Does the introduction provide sufficient background and include all relevant references? |
|
Response: Thank you for pointing this out. We have added more New Zealand references. See the revised introduction.
|
|
3.2 Specific comments |
|
Comments 1: Organizational and social support services are kind of resilience resources and generalized to be the support needs for people living with dementia and their carers. The incentives to conduct this research described in lines 50-61 seemed too weak. |
|
Response 1: Thank you for your comments. We have added an additional paragraph to support the rationale for conducting this research. A new paragraph has been added to Page 2, Line 67-90.
|
|
Comments 2: Reference 12 is identified factors that support people living with dementia and their carers. Why used this study to be the resilience resources framework? |
|
Response 2: Thank you for your question. Reference 12 is Han S, Chi NC, Han C, Oliver DP, Washington K, Demiris G. Adapting the Resilience Framework for Family Caregivers of Hospice Patients With Dementia. Am J Alzheimers Dis Other Demen. 2019;34(6):399-411. doi:10.1177/1533317519862095 The resilience resources framework was adopted from the ecological resilience framework developed by Windle and Bennett (2011), which has been widely used in various studies. However, this framework is not specific to the experience of dementia (Windle et al., 2023). Several studies have used this framework as an initial foundation to further understand and advance knowledge of the resilience of family carers of people living with dementia (Cherry et al., 2013; Donnellan et al., 2017;Han et al., 2019; Joling et al., 2016; Teahan et al., 2018).One such study, by Han et al. (2019), modified this framework for carers of people living with dementia, providing valuable insights into the resources that support people living with dementia and their carers, particularly in the context of hospice care. The application of the resilience resources framework helped us to comprehensively present a variety of resources for resilience at multiple levels. The use of the resilience resources framework in this study helps to conceptualise and operationalise the resilience resources available to Chinese carers of people living with dementia and provides a structured framework for assessing and promoting resilience in this vulnerable population. Therefore, the Resilience Resources Framework appears to be more appropriate for the study of carers of people with dementia.
|
|
Comments 3: Design: A descriptive qualitative method was used to explore the support needs of Chinese family carers of people living with dementia during the COVID-19 pandemic. The rationale for using this method was to understand the individual human experience in its unique context. There are conflicts between lines 65-70 and lines 75-79. |
|
Response 3: Thank you for pointing this out. We acknowledged that these two sentences may cause conflict. This statement has been modified. Descriptive qualitative research focuses on describing and interpreting the experiences, perspectives, and meanings of participants in their natural settings. It allows researchers to capture rich, detailed data about participants' experiences and perceptions. This method is consistent with the aim of this research.
The sentence has been modified in Page 2, Line 97-99. “The rationale for using this method was to understand the individual human perception in its unique context”.
|
|
Comments 4: The process of thematic analysis described in lines 117-118 is an inductive approach. Where is the deductive approach in the data analysis? |
|
Response 4: Thank you for pointing this out. The deductive approach is mentioned in Page 3, Lines 137-139. “The deductive approach was used to map the identified themes onto three levels within the framework of resilience resources.”
|
|
Comments 5: The rigor of this research must be described. |
|
Response 5: We agreed this comment. The rigor of this research is mentioned in Section 2.5 (Page 3, Lines 150-160). The rigor of this study was based on the principle of trustworthiness developed by (Lincoln & Guba, 1985), including credibility, confirmability, dependability and transferability. To enhance credibility and confirmability, the first author met frequently with other authors experienced in qualitative research and carers research to discuss the study design, formulate the analysis strategy, and receive ongoing feedback on preliminary findings. Ensuring dependability of our findings, two interviewers with used the same interview guide. After interviewing each participant, the verbatim manuscripts were analysed and discussed with the other authors. Finally, participants were recruited by a single researcher to increase transferability. Furthermore, the entire interview was tape-recorded to reduce respondent burden and increase data richness.
|
|
Comments 6: The health conditions of the carers might be the confounding factors of the results. |
|
Response 6: Thank you for raising this issue. While we acknowledge that the health conditions of carers could potentially act as a confounding factor, it is important to note that this study used a qualitative research design. The use of purposive sampling may introduce bias in the selection of participants. However, our findings did not show a strong association between carers' health conditions and study outcomes. This could be a limitation of this qualitative research. Given the varied health conditions of carers, it is difficult to make a conclusion that health conditions of carers may be a confounding factor in this study. Therefore, future research should specifically address the health conditions of carers to gain a clearer understanding of its potential influence.
|
|
Comments 7: The definition of resilience resources has to be stated clearly. |
|
Thanks for your comment. There is no single definition of resilience resources in dementia carers research. However, a common understanding is that resilience is shaped by a variety of resources., including individual resources, community resources, and societal resource. Therefore, the operational definition of resilience resources in this study can be defined as carers hold these resources to facilitate or hinder resilience. The resources that family carers possess or lack that either promote or hinder their ability to cope effectively with the challenges of caring for someone with dementia.
The operational definition of resilience resources has been added to the manuscript, which can be found on Page 2, Line 77-78.
|
|
Comments 8:What are the main findings of this research? The contents of Table 2 are not consistent with the text. |
|
Response 8: The main findings of this research were identified the existing and absence resilience sources for Chinese carers. Table 2 and text in the Results have been revised.
The revised information can be found on Page 5, Lines 216-224.
|
|
Comments 9:The authors need to reorganize the research’s findings into one Table. |
|
Response 9: This is an error, it has been deleted.
|
|
Comments 10:What are the meanings of Title1-Title4? |
|
Response 10: This is an error, it has been removed from the site.
|
|
Comments 11:What are the differences between support needs and resilience resources? |
|
Response 11: Thank you for highlighting this distinction. Support needs and resilience resources are two different concepts. Support needs refer to the requirements expressed by carers of people living with dementia who seek additional support to navigate challenging situations. However, resilience resources are specific sources of support or assets that assist carers in coping with and overcoming these challenges. By first identifying the broader term of support needs and then narrowing down to specific resilience resources, interventions can be more effectively tailored to address the diverse needs of carers of people living with dementia. Carers rely on these resilience resources to promote resilience and enable them to cope effectively with the demands of their caring role. A lack of these resources can negatively affect their well-being or exacerbate the challenges they face in their caring role. To address this, it is essential to understand and link their support needs to the availability of resilience resources.
|
|
Comments 12:Discussion: “The available evidence indicates that dementia carers experienced a lack of adequate organizational and community support services during the COVID-19 pandemic, which is consistent with previous findings that the pandemic had a significant impact on the…. “These results are not seen in the text. |
|
Response 12: This information is provided in Table 2 and in the text of the results. This information can be found on Page 5, Lines 216-224.
|
|
Comments 13:lines 378-380: Notably, one-third of absence resilience resources can be interpreted at both individual and community levels, community and societal levels, or individual and societal levels. lines 386-388: This is because the purpose of the framework of resilience resources was to categorize these support needs across the ecological levels rather than differentiate them at each level 12. These results are not seen in the text. |
|
Response 13: We have revised the results and made them more specific. The amendments have been added to the results and Table 2. |
